# Simulating record-shattering cold winters of the beginning of the 21st century in France

Camille Cadiou[1] and Pascal Yiou[1]

[1]Laboratoire des Sciences du Climat et de l'Environnement, UMR 8212 CEA-CNRS-UVSQ, IPSL and U Paris-Saclay, 91191 Gif-sur-Yvette CEDEX, France

**Correspondence:** Camille Cadiou (camille.cadiou@lsce.ipsl.fr)

**Abstract.** Extreme winter cold temperatures in Europe have huge societal impacts. Being able to simulate worst-case scenarios of such events for present and future climates is hence crucial for short and long-term adaptation. In this paper, we are interested in persisting cold events, whose probability is deemed to decrease with climate change. Large ensembles of simulations allow us to better analyse the mechanisms and characteristics of such events, but can require significant computational resources.
Rather than simulating very large ensembles of normal climate trajectories, rare event algorithms allow sampling the tail of distributions more efficiently. Such algorithms have been applied to simulate extreme heat waves. They have emphasized the role of atmospheric circulation in such extremes. The goal of this study is to evaluate the dynamics of extreme cold spells simulated by a rare event algorithm. We focus on winter cold temperatures that have occurred in France from 1950 to 2021. We investigate winter mean temperatures (December, January, and February) and identify a record-shattering event in 1963.
We find that, although the frequency of extreme cold spells decreases with time, their intensity is stationary. We apply a stochastic weather generator approach with importance sampling, to simulate the coldest winters that could occur in a factual and counterfactual climate. We hence simulate ensembles of worst winter cold spells that are consistent with reanalysis data. We find that a few simulations reach colder temperatures than the historical record-shattering event of 1963. This shows that present-day conditions can trigger winters as cold as that record, in spite of global warming. The atmospheric circulation that
prevails during those events is analyzed and compared to the observed circulation during the record-breaking events, showing no main change in the mechanisms leading to this type of extreme events.

## 1 Introduction

Winter cold spells in the mid latitudes have had wide-ranging impacts, affecting agriculture (Trnka et al., 2011; Vogel et al., 2019), health (Gasparrini et al., 2015; Smith and Sheridan, 2019), infrastructures (Chang et al., 2007), or energy systems
(Añel et al., 2017; Bessec and Fouquau, 2008; Van Der Wiel et al., 2019). Cold events are expected to decrease both in terms of intensity and frequency with climate change in most regions of the world (Seneviratne et al., 2021), which could lead to an overall reduction of their impacts. In the last decades a decrease in intensity has been recorded in the Western European region (Cattiaux et al., 2010; Seneviratne et al., 2021; Smith and Sheridan, 2020; Van Oldenborgh et al., 2019). But even if their probability decreases, extreme cold winter events still occur and can cause major disruptions, like winter 2010 in Western

Europe (Cattiaux et al., 2010) or the cold snap of February 2021 in Texas (Doss-Gollin et al., 2021). Winter 2010 was perceived as extremely cold in Europe and rose questions in the media and general public about the occurrence of extreme cold events under climate change. Cattiaux et al. (2010) showed that winter 2010 was actually not as extreme as records of the previous decades and would have been much more extreme given the same atmospheric conditions if it had occurred in a past climate with a lower influence of climate change. This is consistent with the general upward trend of winter minimum temperatures, as shown in Fig. 1a for various lengths of cold episodes.

However, uncertainties remain about the potential dynamical effects of climate change on severe winter mid-latitudes weather (Cohen et al., 2020; Horton et al., 2015; Overland et al., 2016; Shepherd, 2015). The Arctic Amplification (AA) is a mechanism that may lead to an increase in severe winter weather in the mid-latitudes (Cohen et al., 2014; Francis et al., 2018; Francis and Vavrus, 2012; Vavrus, 2018), but its potential effect is intertwined with other hemispheric drivers of decadal variability and the quantification of its influence remains debated (Blackport and Screen, 2020; Cohen et al., 2020; Francis, 2017). Several studies have also shown that winter warming and winter anomalies trends are not as large as the upward trend of summer warm anomalies in the northern hemisphere (Robeson et al., 2014) and more specifically in France (Ribes et al., 2022). The decrease in winter cold spells would consequently not be as significant as the increase in summer heat waves.

The goal of this study is to assess whether we could simulate a winter as cold a 20th century-like cold winter record using present-day climate conditions, and how their intensity and mechanisms are affected by climate change. We aim at building *storylines* of a worst-case winter scenario in France for the present decades (here, "worst-case" scenario is meant to identify conditions leading to events that are colder than already observed). Storylines are "physically self-consistent unfolding of past events, or of plausible future events or pathways". They aim at better understanding the driving factors of high-impact events with deep uncertainties in a way that is more understandable by people than a purely probability-based risk assessment, and exploring the "boundaries of plausibility" (Shepherd et al., 2018).

To assess worst-case scenario winter temperatures in France, we use the winter of 1962-1963 (hereafter referred as winter 1963) as a reference event. Winter 1963 yielded an exceptionally low mean temperature anomaly over December, January and February (DJF) of $-3.4\sigma$ below the seasonal average (Fig. 1). 1963 was also extreme by its spatial scale, with negative temperatures covering most of Europe (Greatbatch et al., 2015; Hirschi and Sinha, 2007; O'connor, 1963). This lead to exceptional weather across the continent: large lakes, like Lake Constance or Lake Zurich, froze entirely and widespread and persistent snow coverage was observed in the British Isles. According to the Met office, it was the coldest winter since 1740 in the United Kingdom. In France, a first intense cold wave occurred at the end of December, lasting one week, followed by a second more prolonged cold wave with negative daily mean temperature over France from the 11th of January to the 6th of February (Fig. 1b). Winter 1963 was associated with a negative North Atlantic Oscillation (NAO) index, indicating lower than normal pressure difference between the Iceland low and Azores high-pressure systems (Cattiaux et al., 2010; Greatbatch et al., 2015). A persistent negative NAO phase is usually associated with the development of North-Atlantic atmospheric blockings (Shabbar et al., 2001), and a weakening of the westerlies allowing outbreaks of cold air coming from the Arctic or Russia into Western Europe (Greatbatch, 2000; Hurrell et al., 2003).

According to the definition of Fischer et al. (2021), winter 1963 was a record-shattering event in France, i.e. the record of low temperatures was broken with a large margin of several standard deviations (Fig. 1a). Therefore, winter 1963 was an extremely low probability event, even considering the colder climate (than in the 21st century) in which it occurred. If the average winter temperature follows a Gaussian distribution, this corresponds to a return period larger than $10^3$ years, which is longer than the observational periods. The objective of this study is to examine whether:

1. a winter as cold as that of 1963 can be generated by a statistical model given information that excludes that specific winter, and

2. such a winter can still be simulated given climate information from recent decades, during which climate has been warmer.

Simulating an ensemble of events whose return period is larger than the observational period, or the typical length of climate model simulations (e.g. $\approx$250 years for simulations from the Coupled Model Intercomparison Project phase 6 (CMIP6) (Eyring et al., 2016), and concatenating "historical" (1850–2014) and "scenario" (2015-2100) simulations) requires intensive computing resources. Even if a large ensemble of simulations is considered (e.g. up to $\approx 50$ ensemble members in the CMIP6 archive), the 2000–2050 period would yield $\approx 50 \times 50$ years of simulations (e.g. with the CESM2 model (Danabasoglu et al., 2020)), which would lead to 2–3 examples of events with a return period of $10^3$ years, such as the winter of 1963. Several methods based on principles of statistical physics have been developed to provide fast and realistic simulations of large values of atmospheric variables. Rare events algorithm using importance sampling (e.g. Ragone et al., 2018) have been designed to specifically simulate extreme heat waves from a climate model. A simplified simulation approach was proposed by Gessner et al. (2021), by carefully selecting trajectories of a climate model that lead to lower temperatures over a given region (Germany). An alternative approach is based on Stochastic Weather Generators (SWG). SWGs are Markov processes used to generate large ensembles of atmospheric trajectories with realistic statistical properties, at a low computational cost (Ailliot et al., 2015). Yiou and Jézéquel (2020) combined an SWG based on analogs of circulation (i.e. days with a similar atmospheric simulation) developed by Yiou (2014) and the importance sampling principle exposed by Ragone and Bouchet (2021) to specifically simulate extreme summer heat waves from analogs of circulation. This method allows for the simulation of an ensemble of physically consistent trajectories of extreme events at a very low computational cost. In this study, we adapt the analog-based stochastic weather generator with importance sampling of Yiou and Déandréis (2019) to the simulation of low-likelihood, high-impact extreme winter events.

We use SWG simulations based on reanalysis data in order to assess the intensity of the worst-case winter scenario in France in a counterfactual period (with lower influence of climate change) and a factual period representing the present-day climate with more detectable influence of climate change. Therefore, we examine how climate change affects the intensity of worst case cold winters that can hit France, and in what extent extreme cold winters like 1963 could still be possible in the present-day climate.

A study by Sippel et al. (2023) has conducted a similar analysis of winter 1963 for Germany, using both ensemble boosting of the CESM2-ETH model and the SWG displayed in this paper. This paper provided an investigation of such winters focusing

on France and a more in-depth analysis of the use of the analogues-SWG. The present paper details the SWG methodology and shows an application for France.

The paper is organized as follows. Section 2 presents the datasets used in the paper and details the analog-based stochastic weather generator model for sampling cold winters. Section 3 shows the results of the simulations of extreme winters. Section 4 discusses the main results of the study.

## 2 Data and Methods

### 2.1 Data

Daily mean surface temperature (TG) data were obtained from the 5th version (ERA5) of the atmospheric reanalysis of the European Centre for Medium-Range Weather Forecasts (ECMWF) (Hersbach et al., 2020). Data from 1950 to 2021 has been retrieved with a spatial resolution of $0.25° \times 0.25°$. Daily temperature fields have been averaged over the smallest spatial domain including metropolitan France (5°W – 9°E ; 42°N – 52°N). ERA5 was chosen for its large time coverage and its high horizontal resolution of $0.25°$.

We compute running averages of temperature for four event durations ($r = 3$, 10, 30 and 90 days) and determine the minimum value for each winter (from December to February). This corresponds to identifying the coldest $r$-day period for each year, or TG$r$d. For an even value of $r$, we compute the minimum over all time steps $t$ of:

$$\mathrm{TG}r\mathrm{d}_t = \frac{1}{r} \sum_{i=-r/2}^{(r/2)-1} \mathrm{TG}_{t+i} \tag{1}$$

where $\mathrm{TG}_t$ is the daily average temperature at time step $t$. For an odd value of $r$, the equivalent sum is centered around $r$. Fig. 1a shows the variations of TG$r$d time series for France. Additional information on the associated distributions is shown in Appendix A. We observe an upward trend of TG$r$d at the four event durations $r$. The records at each event duration occurred before 1990. However, extreme cold events still happened in the 21st century: winter 2012 witnessed the 5th coldest 10-day and the 8th coldest 3-day cold spells of the 1950-2021 period. At the 30-day event duration, February 1956 and January 1963 were two very extreme events, with temperature anomalies to the 1950-2021 trend of respectively $-2.9\sigma$ and $-2.8\sigma$ (see dashed black lines in Fig. 1a). One event stands out in the 90-day winter temperatures: winter 1963 is at $3.7\sigma$ from the trend of winter mean temperatures and $2.2\sigma$ under the second coldest winter in 1950-2021, winter 1952-1953. For conciseness, this paper focuses on TG90d, i.e. DJF average temperatures. This assumption of the distribution of TG$r$d to be Gaussian is experimentally justified in Appendix A.

For the computation of analogs of circulation, we use daily geopotential height at 500 hPa (Z500) from ERA5 reanalysis from 1950 to 2021. Z500 was chosen over SLP because of its lower sensibility to perturbations from the surface roughness and its common use on weather regime studies (Corti et al., 1999; Yiou and Nogaj, 2004; Jézéquel et al., 2018; Dawson et al., 2012). Jézéquel et al. (2018) also showed it was better suited to simulate temperature anomalies, although that study investigated warm temperatures. Z500 data were regridded on a $1° \times 1°$ grid to reduce computation time since a higher resolution has little impact

on the analogs calculation because of the smooth spatial variability of Z500 fields. We considered the Z500 field over the North Atlantic region (20°W – 30°E ; 30°N – 70°N) to compute circulation analogs. This domain offers a compromise between a spatial coverage large enough to study the role of the synoptic circulation but small enough not to drown out the signal in the too complex hemispheric circulation.

## 2.2 Analogs of circulation

We first compute a database of analogs of circulation following the procedure of Yiou and Jézéquel (2020). For a given day $t$, we compute the Euclidean distance of the Z500 fields between $t$ and all days $t'$ that are not in the same year or season (astride two following years) and within a calendar distance to $t$ inferior to 30 days, (i.e. the difference in days between two days, regardless of the year). The $K$ analog days of $t$ are the $K$ days for which the distance from $t$ is the smallest. We chose $K = 20$ analogs, as advocated in previous studies (Platzer et al., 2021).

The circulation analogs were computed using the "Blackswan" Web Processing Service (Hempelmann et al., 2018). We consider three different analog data sets, depending on the time period in which the analogs are selected:

1. 1950-2021: the whole length of available ERA5 data,

2. 1950-1999: the past period, as a counterfactual, with less influence of anthropogenic climate change.

3. 1972-2021: the present period, as a factual, with a significant signal from climate change.

The analog periods were chosen to have sufficient length to be representative of all the possible states of the atmospheric patterns while being characteristic of climate periods that are significantly different. We were constrained by the length of the ERA5 data available (only 71 years). Therefore we chose a compromise between analog depth and low overlap by considering two 50-year periods (1950-1999 and 1972-2021). The distribution of analog years and their quality is detailed in Appendix B

## 2.3 Stochastic Weather Generator and importance sampling

The analog-based stochastic weather generator (hereafter referred as SWG) developed by Yiou (2014) uses stochastic reshuffling of daily atmospheric fields to generate atmospherically-consistent alternative trajectories of climate events. This algorithm was adapted by Yiou and Jézéquel (2020) to simulate extreme heat waves using a principle of importance sampling. The goal is to simulate $L$-day trajectories of a model while maximizing an observable (e.g. local temperature). Here we focus on a modified version of the so-called "dynamic" type of simulations, developed by Yiou (2014), which computes alternative atmospheric trajectories starting with the same initial conditions as an observed event. Each time step is the combination of an atmospheric circulation (characterized by Z500) and a variable of interest, such as temperature. The SWG starts at a given initial condition and goes from one time step to the following using analogs of circulation according to the process described hereafter. The resulting simulation is a constrained reshuffling of days of the input dataset. We start at an initial condition $t_0$, which thus constitutes the first time step of the simulation. To simulate temperature for the day after $t_0$ ($t = t_0 + 1$), we randomly pick one analog among the $K$ best analogs of the observed Z500 field on day $t_0 + 1$. The geopotential height at 500hPa (Z500) and the

2m temperature (TG) fields of this analog day constitute the simulated day $t'$. For the next time step, $t$ is replaced by $t' + 1$, the day following $t'$. This random process is repeated sequentially for $L$ time steps, the length of the simulation. This defines a Markov chain of TG, with latent states provided by the analogs of Z500.

At each time step, the selection of the analog day follows several constraints and weights controlled by the parameters that are described hereafter. To better follow the seasonal cycle of the simulated season, we use $K$ weights $\omega_{cal}^{(k)}$ ($k \in \{1, \ldots, K\}$)

on the analog selection that depend on a parameter $\alpha_{cal}$ which favours analog days that are closest to the calendar date (i.e. day of year, regardless of the year) of time step $t$ :

$$\omega_{cal}^{(k)} = A_{cal} e^{-\alpha_{cal} d_k} \tag{2}$$

where $A_{cal}$ is a normalizing constant, $\alpha_{cal} \geq 0$ is the calendar weight and $d_k$ is the number of calendar days between the $k^{th}$ analog day and $t$. The resulting seasonal cycle of the SWG is shown in Appendix C.

To favour the simulation of the most extreme events, importance sampling weights $\omega_T^{(k)}$ are introduced, with a control param-

eter $\alpha_T \geq 0$. When the value of $\alpha_T$ increases, the stochastic weather generator favours analog days with extreme temperatures. The $K$ analogs of $t$ are sorted in ascending order of temperature with ranks $R_k$ ($k \in \{1, \ldots, K\}$), so that the coldest analog day has a rank of 1. Hence the weight associated to the $k$th analog day of $t$ is:

$$\omega_T^{(k)} = A_T e^{-\alpha_T R_k} \tag{3}$$

where $A_T$ a normalizing constant, $\alpha_T$ is the importance sampling weight and $R_k$ is the rank (in ascending order) of the $k$th analog day among the $K$ analogs. If $\alpha_T = 0$, there is no importance sampling and the SWG is the same as in (Yiou, 2014).

As explained by Yiou and Jézéquel (2020), using the ranks of temperature instead of their absolute values eliminates the need to re-scale variables, and allows for an interpretation independent of the units of the variables. It also ensures that simulations are biased in the same way, as importance sampling weights are the same at each time step (the ranks are simply all integers between 1 and $K = 20$).

The SWG with importance sampling is obtained by combining the weights on the calendar and importance sampling. The

$k$th analog day of day $t$ has a probability of:

$$\omega_k = A e^{-\alpha_{cal} d_k} e^{-\alpha_T R_k} \tag{4}$$

where $A$ is a normalizing constant so that $\sum_{k=1}^{K} \omega_k = 1$.

### 2.4   SWG configurations

We have refined the original methodology proposed by Yiou and Jézéquel (2020) by introducing a new approach for simulating trajectories with the analogs-based SWG. Our primary objective is to generate extreme winter events that are characteristic

specific climate periods, essentially seeking to estimate the extremes attainable within a climate state characterized by a defined level of warming. Given a reference record-shattering event, we want to assess the likelihood of a similar event occurring within a given climate period, without specific information about the event itself.

To achieve this objective, we have made a crucial adjustment of the original SWG configuration (configuration 1, which considers the entire data in the simulations). This new configuration (configuration 2) no longer considers analogs that coincide with the reference event. At each time step, the selection process is based on the $K$ best analogs, but the weights $\omega_k$ of analog days that fall within the observed event are set to zero. For example, if we initiate a simulation on December 1st, 1962, spanning $L = 90$ days, all analog days falling between December 1st, 1962, and March 1st, 1963, are excluded from consideration in the simulation. Days of the same year were already excluded during the analog computation process, but they can still be selected in simulations as analogs of days in other years. This procedural adjustment ensures that our simulations are solely driven by the initial conditions and the state of the climate, and do not rely on any specific information pertaining to the observed event (apart from the first day). In other words, we are assessing whether a record-shattering event of 90 days can be inferred from information related to less extreme events.

We also adapted this stochastic weather generator from the one of Yiou and Jézéquel (2020), which was designed to simulate summer heat waves, to the version used in this analysis tailored for simulating winter cold spells. Consequently, the importance sampling weights now favour the coldest analogs instead of the warmest ones.

## 2.5 Consistency of SWG trajectories

To control the consistency of the atmospheric trajectories produced by the SWG, we compute the derivatives of the resulting time series of Z500 and temperature from first-order differences. The derivatives are compared on the one hand to the derivatives obtained in observed winter, and on the other hand to time series computed by randomly picked winter days in the ERA5 dataset. The results are shown in Fig. 2a and b. Because of the approximation by analogs, the SWG loses continuity, which appears in the derivative calculation. But the ratio of Z500 derivatives standard deviation between the simulations and the observations is $\approx 1.2$, which is far less than the same ratio computed from random time series ($\approx 3.5$). This illustrates that the SWG atmospheric trajectories are mathematically consistent and realistic. The same analysis for daily temperature derivative shows a larger difference between the ERA5 reanalysis and the SWG simulations. This can be explained by the fact that the analogs-SWG trajectories are primarily based on an atmospheric trajectories constraint (and not a constraint on temperature). The resulting temperature time series are still more consistent compared to time series of random winter days (the ratio of standard deviation derivatives with the reanalysis of respectively $\approx 1.8$ and $\approx 3.2$). Therefore, the short-term variability (i.e., a few days) is deemed to be relevant from the mathematical point of view.

Finally, we compare the mean temperatures of 1000 SWG trajectories with $\alpha_T$ set to 0.5 to the mean temperatures of $10^7$ control SWG trajectories with $\alpha_T$ set to 0 (i.e. without importance sampling) and with analogs in 1950-1999 (Fig. 2c). In the control simulations, the SWG covers the range of observed winter TG90d, apart for very extreme values in the tail of the distribution, as for winter 1963. The median of simulations with importance sampling is close to the coldest control simulation. The SWG with importance sampling simulates very extreme events in the tail of the distribution, and is able to reach values of the same order of magnitude as winter 1963. The difficulty to obtain winters as cold as 1963 without importance sampling shows the added value of this method for record-shattering events. The resulting distributions are comparable to the ones

obtained for heat waves with a rare event algorithm by Ragone and Bouchet (2021), and a control simulation of a climate model.

We verify in Appendix B that the selection of analogs does not yield obvious biases towards the earlier parts of each analog period, and that the quality of analogs is stationary in time across the two periods. Appendix C compares the climatological variations of the SWG without importance sampling with the original ERA5 data.

## 2.6 Protocol

First we tune the $\alpha_T$ and $\alpha_{cal}$ parameters for winter DJF simulations. We simulated ensembles of trajectories of 90 days starting on Dec. 1st of each year, with different values of parameter $\alpha_{cal}$. Figure 3a shows the percentage of simulations for which the last day of the simulations falls after the 15th of February. If the calendar weight value is too small (e.g. $\alpha_{cal} = 1$), fewer than half of the simulations end with a calendar day after the 15th of February. This means that the trajectories of the simulated events with $\alpha_{cal} = 1$ are less consistent with the seasonal cycle. With an $\alpha_{cal}$ parameter greater than 5, more than 75% of the simulations have their last day falling after the 15th of February. Therefore we use $\alpha_{cal} = 5$ in the following.

We run the SWG with parameter values $\alpha_T \in \{0, 0.1, 0.2, 0.5, 0.75, 1\}$, starting on each 1st of December, between 1950 and 2021. Figure 3b shows that for $\alpha_T = 0$, the SWG simulates events covering the range of winter mean temperatures from 1950-1951 to 2020-2021, apart from winter 1962-1963. For $\alpha_T = 0.2$, a few outlier simulations reach winter 1963 temperatures. A value of $\alpha_T$ greater than 0.5 allows the simulation of a greater proportion of extreme events. The difference for $\alpha_T$ greater than 0.5 being less significant, we chose for the following $\alpha_T = 0.5$.

To compare the configuration of the SWG excluding information from the event of reference from the previous configuration of (Yiou and Jézéquel, 2020), we use both to simulate worst-case winter scenario from 1950 to 2021. For each winter from 1950 to 2021, the simulation starts at $t_0$ — the 1st of December — and run for $L = 90$ days over DJF (December, January and February). $n = 100$ simulations are run by winter year, hence $100 \times 71$ events are simulated for each experiment.

Then we focus on the record-shattering event of winter 1963. Winter 1963 being our reference as a record-shattering event, we simulate alternative extreme winters in the counterfactual and factual climates without using the information from winter 1963 to evaluate to what extent such a winter can be extrapolated from available data in both climate periods. Simulations are also made using the two sets of analogs — the counterfactual and factual periods. Hence we simulate winters that could have occurred in the selected analog period. This allows simulating worst case events from the same initial conditions but considering different climate states. We run $n = 1000$ simulations starting in December the 1st 1962, using as previously, the factual and counterfactual sets of analogs. This allows having a wider ensemble of possible winter temperatures starting from the initial conditions of winter 1963.

## 3 Results

### 3.1 Sensitivity to SWG configurations

In this subsection, we simulate cold winters of 90 days starting on a 1st of December of each year from 1950 to 2021. Analogs can be selected in any year from 1950 to 2021. We evaluate the impact of the possibility to sample analogs from the reference event on the winter average temperature: the SWG configuration of Yiou and Jézéquel (2020), versus the one of the present paper.

Figure 4 shows the results of simulations from 1951 to 2021 using respectively SWG configuration 1 (Fig. 4a) and SWG configuration 2 (Fig. 4b) in Section 2, with analogs sampled in 1950-2021. The SWG successfully simulates extremely cold winters, with simulations being respectively 3.9°C (Fig. 4a) and 3.1°C (Fig. 4b) colder overall compared to the long-term mean of ERA5 temperatures. 40% of all simulations reach a mean temperature as cold as the 1963 record with configuration 1, while only 13% of them are as cold using configuration 2.

The variability of the simulations performed with configuration 1 follows closely the variability of historical winter temperatures due to the possibility of selecting analogs falling in the observed event with this configuration. The medians of simulations made with configuration 1 are highly correlated ($r = 0.88$) to the observed temperatures. With configuration 2 there is no correlation between the median of simulations and the observed temperature. This is the consequence that configuration 2 does not use information from the observed event apart from the initial conditions. The simulation length of 90 days being longer than the decorrelation time of atmospheric dynamics, the resulting events should not be highly influenced by their initial conditions. The standard deviation of the medians of the boxplots obtained with configuration 2 is also very low: $0.15$°C, compared to the $0.44$°C obtained with configuration 1. This is coherent with the chaotic internal variability of the climate system, resulting in simulated events being representative of the climate analogs period used, rather than the initial conditions.

The mean of boxplot median is also higher by $0.77$°C with configuration 2 compared to configuration 1, which can be explained by the fact that configuration 1 allows more days to be selected during the importance sampling process, so that the coldest days can be selected during the simulations by construction, while some analog days are excluded when using configuration 2.

### 3.2 Focus on winter 1963

In this subsection, we simulate winters, starting on Dec. 1st 1962, and consider circulation analogs in the counterfactual and factual periods. We simulate 1000 winters that "could have been" in 1962/1963, with Z500 analogs in two periods. Figure 5a focuses on those simulations for both the factual and counterfactual analogs periods. The first quartile of simulations does not reach 1963 mean values both in the factual and counterfactual periods (boxplots in Figure 5a). 1963 was already a very rare event in the 1950-1999 climate but remains reachable using only 1971-2021 analogs, in a climate with more global warming. The observed increase in mean winter temperatures between the counterfactual and factual periods is not reproduced if we consider extreme winters as simulated by the SWG, even if we exclude winter 1963 from the counterfactual period (configuration 2). We observe an increase of 0.44°C in TG90d between the two periods, while the difference is of only 0.13°C

between the two ensembles of extreme winters. In other words, extremely cold winter do not warm at the same rate as winter mean temperatures.

Figure 5b summarizes the time series of daily average temperatures associated with winter 1963 simulations. The temperatures of winter 1963 are below the seasonal cycle for most of the season. The temperatures of simulated events for both the factual and counterfactual periods are also overall below the seasonal mean for the all length of the event. The median fluctuates around 2°C, while the 5th percentile of all simulation reaches −5°C during most of the winter. The 95th percentile stays under 2°C above the seasonal average, while the 5th percentile reaches −4°C during most of the event, which corresponds to the coldest daily mean temperatures observed in 1963. Overall the range of daily winter temperatures as simulated for extreme winter temperatures in the factual and counterfactual periods matches the range of daily mean temperatures observed in the reference event in 1963. Hence a winter like the one of 1962-1963, even with a low probability, can still be reached using data from the warmer climate of the 21st century. We find that the likelihood of reaching such cold winters barely decreases with time, as the distributions of simulated temperatures are very similar for the two analog periods.

### 3.3 Atmospheric dynamics during winter 1963

During winter 1963, a strong and persisting anticyclonic anomaly prevailed over Iceland. It was associated with a negative Z500 anomaly over continental Europe, the Azores and the Glacial Arctic Ocean, leading to a weakening of the westerlies and advection of cold air from the Arctic (Fig. 6a).

Here, we compute the composites of Z500 and Z500 anomalies over a region that is larger than the region for which the analogs are obtained. The Z500 composite over DJF does not correspond directly to a North Atlantic weather pattern (negative phase of the North Atlantic Oscillation (NAO)), as for instance obtained by Cattiaux et al. (2010). The low over Europe is located more to the East than for an NAO− weather pattern, while the positive Z500 anomaly over Iceland is located more to the north than it would be in an Atlantic ridge weather regime. The respectively positive and negative Z500 anomalies over Iceland and the Azores are however characteristic of NAO− which is often, even if not systematically, an indicator of colder than usual winter temperature over Europe (Hirschi and Sinha, 2007).

The Z500 anomalies of SWG simulations for winter 1963 are spatially smoother than the ERA5 field for the same winter. This can be explained by the fact that the map is averaged over 100 (10% of 1000 simulations) different simulations and that simulations are less auto-correlated than observed events would be, thus having more spatial variability. Hence, we compute the Z500 and Z500 anomaly composites of SWG simulations for the 10% coldest members of the ensemble starting on Dec. 1st 1962. For comparison purposes with winter 1963, this selection is reasonable because 75% of simulations are warmer than this record event (Fig. 5a), and we want to focus on the coldest members of the ensemble. Appendix D also shows that average maps are representative of individual events. We find that the pattern of a strong negative anomaly over Western Europe and a positive anomaly over Iceland is still reflected in the 10% coldest events simulated with the SWG in both the counterfactual (Fig. 6b) and factual (Fig. 6c) simulations. The intensity and position of the Z500 low over the Barents sea and the Z500 high over western Russia seen in ERA5 seem to have a lower contribution to the intensity of the event, as they are weaker and less marked in the SWG simulated events.

## 4 Conclusions

This paper presents the application of an analog stochastic weather generator to simulate ensembles of extreme cold winters in continental France. We adapted the method developed by Yiou and Jézéquel (2020) (to simulate extreme heat waves) to the simulation of extreme cold events. In particular, this paper explicitly addresses the question of simulating the most extreme winter without using information from the observed record. The paper displays a proof of concept using ERA5 data for the simulation of extreme winter temperatures in France between 1950 and 2021.

The SWG for the simulation of extreme cold spells inherits some of the technical caveats already pointed out by Yiou and Jézéquel (2020) for the simulation of extreme heat waves. This SWG method is limited by the length of the data set used as input, so that it may not sample completely the atmospheric dynamics of the climate system. The average of resampled analogs is however bounded to a lesser extent and can reach values far more extreme than the most extreme ones in the input data set. The SWG allows simulation of extreme events outside the observed range, but is still limited by the duration of available data.

The length of the factual and counterfactual periods considered was a compromise between the length of available data (70 years), the non-stationarity of temperatures and the overlapping of the two periods. We needed periods of sufficient length to sample correctly the climate considered. But the shorted overlap between the two periods was necessary to investigate the significance of differences between the factual and counterfactual periods The non stationarity of climate also means that the longer the periods, the less homogeneous they are in terms of level of warming. 50-year periods yield good results in terms of both extremeness (the data set is large enough to simulate very cold winters) and significant enough difference between the factual and counterfactual periods. Moreover we verified that the analog days are evenly distributed over the two climate periods and evenly picked during the simulation process. Therefore we consider that the events simulated are representative of the entire analogs period used in the SWG.

This method does not allow to disentangle anthropogenic warming from others forcings and natural multi-decadal variability of the climate system. But it gives an estimation of the worst-case winter temperatures scenario for a given climate period. Another caveat is that the method is mainly based on the use of flow analogs to assess temperatures. It focuses on the link between atmospheric circulation and temperatures and does not take into account other drivers and feed-backs. For instance, snow cover is not considered in the simulations even though it can have a significant impact on extreme winter temperatures (Orsolini et al., 2013).

We showed that winter as cold as the record event of 1963 or even colder could still occur in the current climate, at a higher level of warming. This does not mean than such an event will happen in the near future but it remains possible at the considered level of warming and is relevant from an adaptation point of view. A winter as cold as 1963 would indeed have major impacts on society, especially on the energy system (Añel et al., 2017). For instance, Doss-Gollin et al. (2021) showed that the February 2021 Texas cold snap, which resulted in major failures of the energy system causing energy, food and water shortages, was actually not unprecedented both in terms of temperature anomalies and resulting heating demand per capita. The lack of preparedness and greater exposure of the energy system due to increasing population and electrification led to disproportionate impacts. In France, the electricity transmission system operator RTE (Réseau de Transport d'Électricité)

estimates the sensitivity of electricity consumption to temperature to be 2400 MW/°C in winter (RTE, 2021). Hence, it might be desirable that energy systems and logistics are scaled for worst case winter scenario in the current or future climate conditions and exposure as the ones simulated in this study.

The possible occurrence of unprecedented cold winter temperatures in France as simulated in this paper is not inconsistent
with the already observed decrease in cold spells intensity in the northern mid-latitudes as exposed by Van Oldenborgh et al. (2019). We focus on very low-likelihood winters, with a return period of over $10^3$ years. This is not representative of meteorological cold waves. For instance, a cold wave in France is defined by Météo France when the national thermal indicator goes under under $-2°C$ for at least three days (Météo France, 2020). A decrease in the intensity of shorter events or an increase in the mean of cold spells is not contradictory to the slow increase in extreme long-lasting winter temperature.

The absence of significant changes in the atmospheric circulation leading to the extreme winters simulated is in line with the *typicality* of large and persistent temperature anomalies as shown using large deviation theory (Galfi and Lucarini (2020); Gálfi et al. (2021)). The same atmospheric conditions usually lead to the most extreme events. However these results are valid in a stationary system and obtained using steady state model simulations. Climate change can lead to important shifts in atmospheric dynamics that could affect the frequency and intensity of extreme events, as well as the dynamics leading to them.
The present paper shows no significant shift in the atmospheric circulation of record-breaking winters between the factual and counterfactual periods, which have a difference $0.72°C$ in terms of level of warming. However these results cannot be extended to a higher shift in global warming level. Simulations of extreme cold spells using of the Coupled Model Intercomparison Project phase 6 (CMIP6) simulations (Eyring et al., 2016) would therefore be an extension of this study in order to further explore the evolution of extreme winter temperatures in the mid-latitudes in the future — according to different emissions
pathways (Riahi et al., 2017) — and the associated atmospheric trajectories.

In this paper, we focused on cold winters (90 days: TG90d) in France. The method can be adapted to simulate cold events of different duration, or in other regions. The worst cold spells recorded in France were February 1956 — the coldest month of the $20^{st}$ century (Andrews, 1956; Dizerens et al., 2017) — and January 1985 (Météo France, 2022a, b). These events caused major disruptions and had a wide health impact (Huynen et al., 2001). The energy sector has been sensitive to 15-day events.
The cold spell of 3rd – 17th January 1985 is used as the event of reference by the French electrical network company. A similar event triggered an unprecedented impulse of solidarity for the help of the homeless during winter 1954.

Winter 1963 was the coldest winter recorded in France and a record-shattering event. Using an analog stochastic weather generator with importance sampling for the simulation of an extremely cold winter, we show that winter 1963 temperatures were already exceptional in the lower level of warming in which it occurred. Estimations of the possibility of such an extreme
event occurring in the recent climate show that it is still possible to have a winter as cold as in 1963, even if it would remain a highly exceptional event. This paper hence provides a *storyline* for extremely cold winters in France (Sillmann et al., 2021).

*Code and data availability.* The ERA5 reanalysis data are publicly available at https://cds.climate.copernicus.eu/. The Ana-SWG code, the processed temperature time series and the analogues files are available on github at https://doi.org/10.5281/zenodo.10726791.

## Appendix A: Probability distributions of TG$r$d according to $r$

We compute the distribution of the annual minima of TG$r$d for the four $r$ values considered in the article (3, 10, 30, 90 days). We obtain empirical probability distributions of TG$r$d in Fig A1. As anticipated, the variance of TG$r$d decreases with increasing $r$. The probability distribution for $r = 90$ (i.e., the distribution of yearly winter mean temperatures) yields a p-value lower than 0.05 when conducting a Shapiro-Wilk normality test, which indicates that it follows a Gaussian distribution. As for the other scales, we compute the minimum of TG$r$d over a yearly block. The resulting distribution should tend to a Generalized Extreme Value (GEV) distribution for lower $r$ (Coles, 2001). However, given the non-stationnarity of the data and the relatively small sample sizes ($n = 71$), it is challenging to draw any definitive conclusion for lower values of $r$.

## Appendix B: Analog quality

ERA5 data is not stationary, because of climate change between 1950 and 2020. This is illustrated for temperature time series at different event durations in Fig 1. This also appears on Z500 variations due to thermal expansion of the atmosphere. Thus we first checked that the non-stationarity does not affect the distribution and quality of the analogs. Fig B1a shows the year of the 20 best analogs of each winter day of 1950-2021, either in 1951-1999 or 1972-2021. Fig B1b compares the quality of the same two sets of analogs (through the value of the Euclidean distance). We verify that there is no major difference between the two periods and that analog years show no meaningful trend. Regarding the analogs selected during the SWG simulations, there may indeed be a slight bias towards earlier years in the dataset, but this is not meaningful. The median and mean of the selected analog years are 1972 and 1973, respectively, for the 1951-1999 period, and 1992 and 1994, respectively, for the 1972-2021 period. Therefore, we conclude that the SWG simulations are representative of the climate period of the analogs.

## Appendix C: Climatology of the SWG

The purpose of the $\alpha_{cal}$ weights is to ensure that the simulations go forward in time and adhere to the seasonal cycle, thereby generating realistic events that do not resemble an "eternal winter" scenario with persistently extreme cold temperatures. Yiou and Jézéquel (2020) developed this calendar weighting approach to achieve this objective, and the proportion of days falling at the end of the period of the event is a method they previously proposed to control how simulated events do not deviate from the seasonal cycle. A climatology of the SWG is implicitly computed in Fig. C1, when we initialize the SWG from all 1st of December (1951 to 2021) in the ERA5 reanalysis with a calendar constraint but without importance sampling (i.e. $\alpha_T = 0$). Seasonal cycles are not smoothed. The figure shows that the SWG reproduces a seasonal cycle, although the mean seasonal cycle is not as cold as in ERA5. This indicates that the simulated winters yield realistic variations around the seasonal cycle, but also that the SWG results are conservative and that colder events may be possible.

## Appendix D: Maps of individual SWG events

Here, we empirically verify that simulations initiated from different conditions yield similar atmospheric patterns. Fig. 6 presents the average composite maps of the 10% coldest SWG simulations for each period. Fig. D1 demonstrates that these average maps are representative of individual events and that the simulated individual extremes tend to resemble each other. For instance, the Z500 mean maps of the 9 coldest simulations from the SWG using analogues from 1972-2021 are shown.

*Author contributions.* CC and PY conceived the experiments from the original code of PY. CC produced the numerical experiments and analyses. Both authors contributed to writing the manuscript.

*Competing interests.* The authors declare that they have no known competing financial interests or personal relationships that could have appeared to influence the work reported in this paper.

*Acknowledgements.* The authors acknowledge the support of the grant ANR-20-CE01-0008-01 (SAMPRACE: PY, CC). This work also received support from the European Union's Horizon 2020 research and innovation programme under grant agreement No. 101003469 (XAIDA: PY).

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

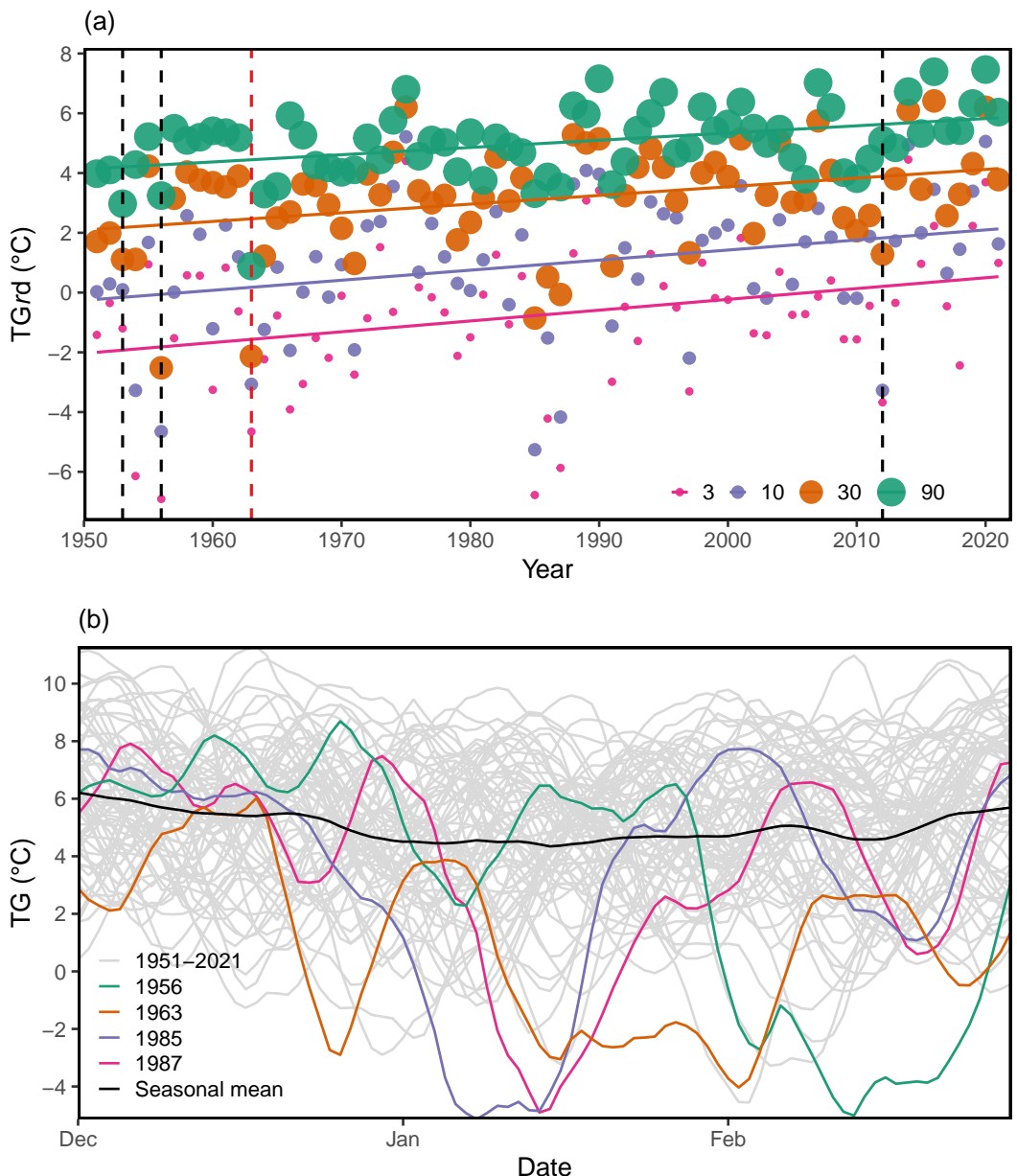

**Figure 1.** **(a)** Minimum of winter temperature over DJF for four event durations in continental France. For each winter we compute the $n$-day running mean of daily mean temperature (TG) for $n \in \{3, 10, 30, 90\}$ days and select the minimum value. The colored lines are linear regressions of the temperature averages. The red vertical dashed line outlines 1963 (the coldest winter in France). The black vertical dashed lines outline the winters of 1953, 1956 and 2012. **(b)** Time series of DJF 7-days running mean TG from 1950 to 2021 in continental France. The curves in colour are for the four coldest years (1956, 1963, 1985 and 1987) and the black curve is for the seasonal mean computed over 1951-2021.

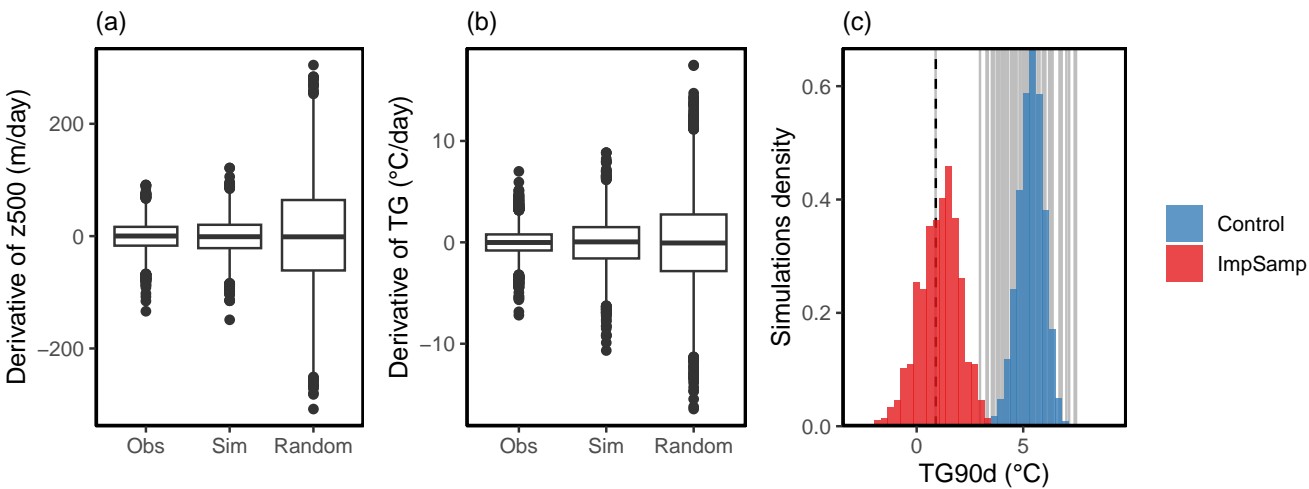

**Figure 2.** Daily derivatives of Z500 (in m/day) **(a)** and TG (in °C/day) **(b)** for winters in ERA5, in the SWG simulations and in random trajectories. **(c)** Normalized distribution of the TG90d of 1000 SWG simulations with $\alpha_{cal} = 0.5$ (red histogram, with importance sampling) and $10^7$ SWG simulations with $\alpha_{cal} = 0$ (blue histogram, without importance sampling). The grey vertical lines display the observed TG90d in ERA5. The black dashed line outlines winter 1963 value.

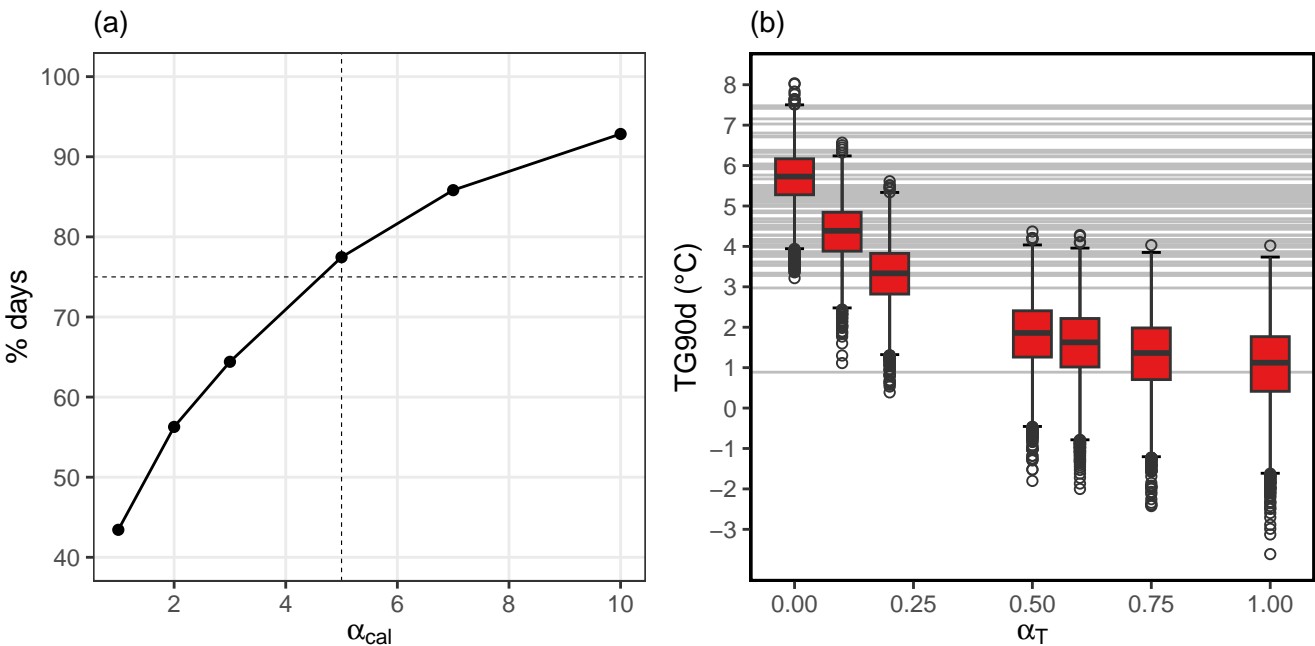

**Figure 3. (a)** Percentage of simulations of extreme winters for which the last day falls after the 15$^{\text{th}}$ of February, as a function of the $\alpha_{cal}$ parameter. **(b)** Temperature (TG) distribution of 100 simulations (configuration 2) for each winter between 1950-1951 and 2020-2021 ($100 \times 70 = 7000$ simulations total per box-plot) done for various values of the $\alpha_T$ parameter ($\alpha_T \in \{0, 0.1, 0.2, 0.5, 0.75, 1\}$). Horizontal lines represent the winter mean temperature of each winter from 1950-1951 to 2020-2021 in ERA5 data. Boxplots are computed as in Fig. 3b.

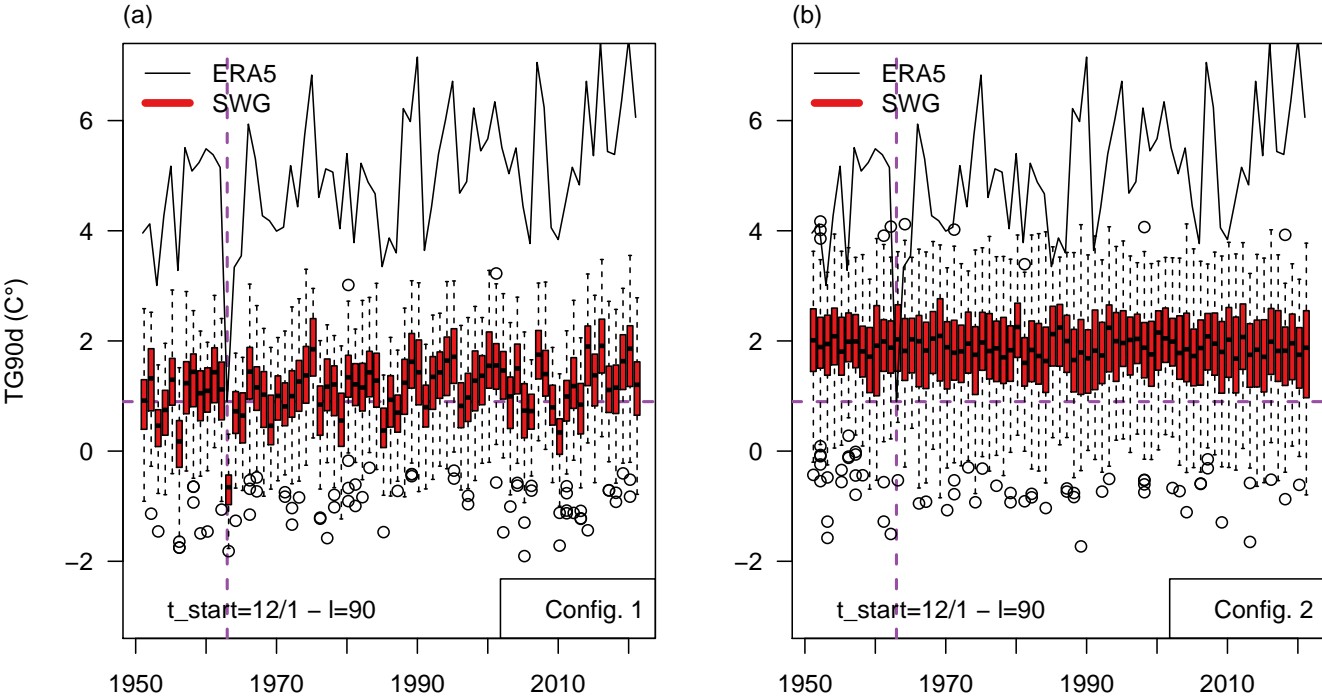

**Figure 4.** Results of 100 SWG simulations from winter 1950-1951 to winter 2020-2021 with configuration 1 **(a)** and configuration 2 **(b)**. The black continuous line represents the time series of winter mean 2m-temperature over France from ERA5 data. The box plots represent the ensemble variability of the simulations for each year. The boxes of box plots indicate the median ($q50$), the lower and upper hinges indicate the first ($q25$) and third ($q75$) quartiles. The upper whiskers indicate $\min[\max(T), q75 + 1.5 \times (q75 - q25)]$. The lower whisker has a symmetric formulation. The points are the outlying values that are above or below the defined whiskers. The vertical purple dashed line highlights winter 1963 while the horizontal purple dashed line shows the mean temperature of the same winter.

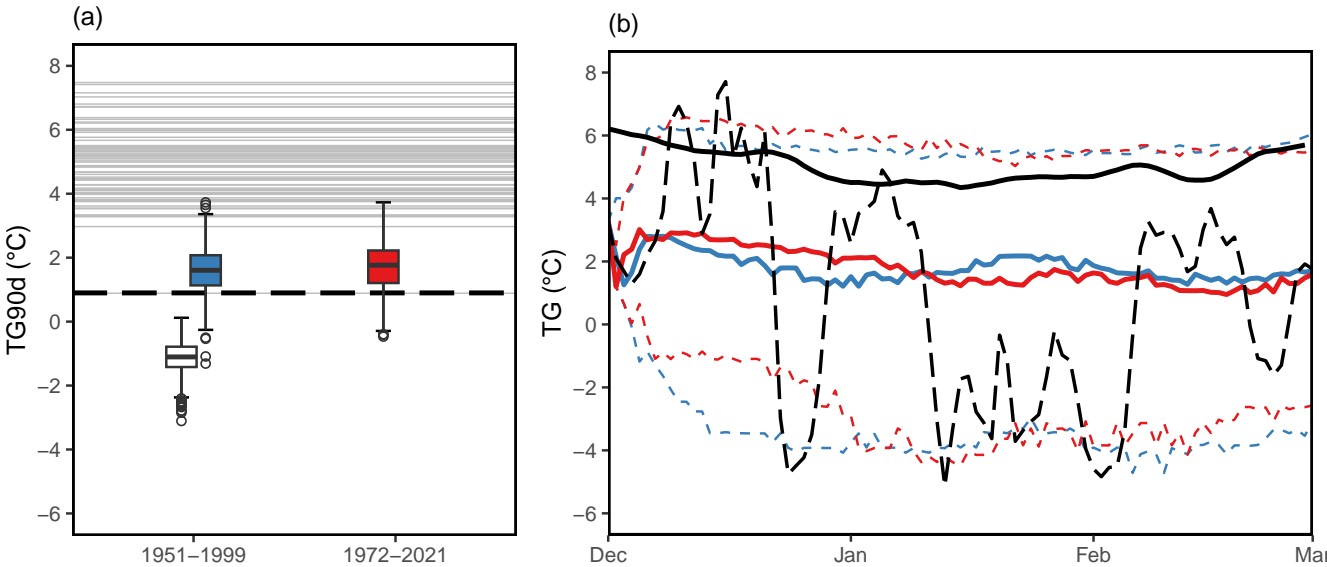

**Figure 5. (a)** Temperature (TG) distribution of 1000 SWG simulations of winter 1963 with analogs from 1950-1999 (left) or 1972-2021 (right) in ERA5 data using configuration 1 (white boxplots) and configuration 2 (color filled boxplots). Horizontal lines represent the winter mean temperature of each winter from 1950 to 2021 in ERA5 data. The dashed black line is the value that was observed in 1962-1963. **(b)** Time series of 7-day running mean of daily mean temperatures for winter 1963 (dashed black line), SWG median (plain line), q05 and q95 (dashed lines) temperatures of the thousand 1962-1963 simulations with configuration 2 using 1950-1999 analogs (blue lines) and 1972-2021 analogs (red lines), and seasonal cycle as computed from 1950-2021 (plain black line).

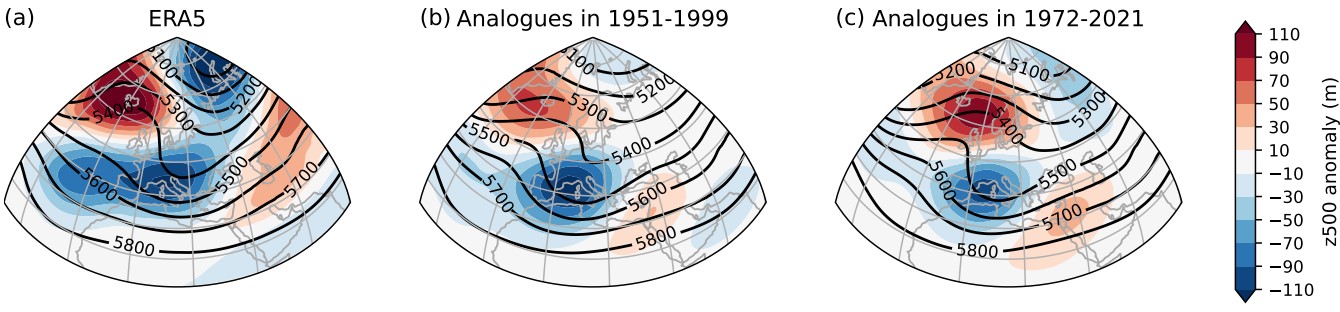

**Figure 6.** Absolute values (contours, in m) and anomalies (in m) with respect to 1950-2021 (shaded areas) of 500-hPa geopotential height (Z500) average over DJF for winter 1963 as observed in ERA5 (a) and simulated by the SWG with counterfactual (b) and factual (c) analogs using configuration 2 (*sine data*). For the simulations, the composite maps are computed from the 10% coldest simulations among the 1000 (i.e. 100 simulations per map).

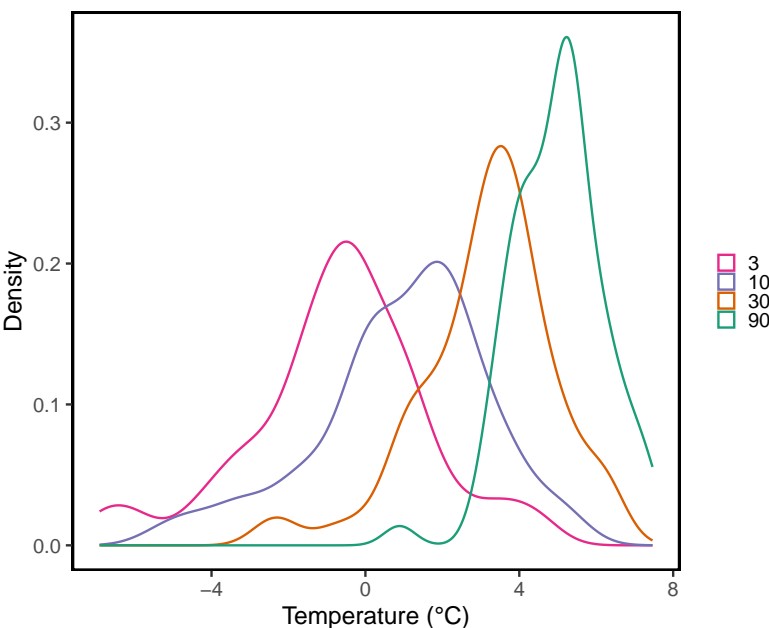

**Figure A1.** Empirical probability distribution functions of TG$r$d temperatures with $r \in \{3, 10, 30, 90\}$ days. The probability density functions are obtained from TG in the ERA5 reanalysis.

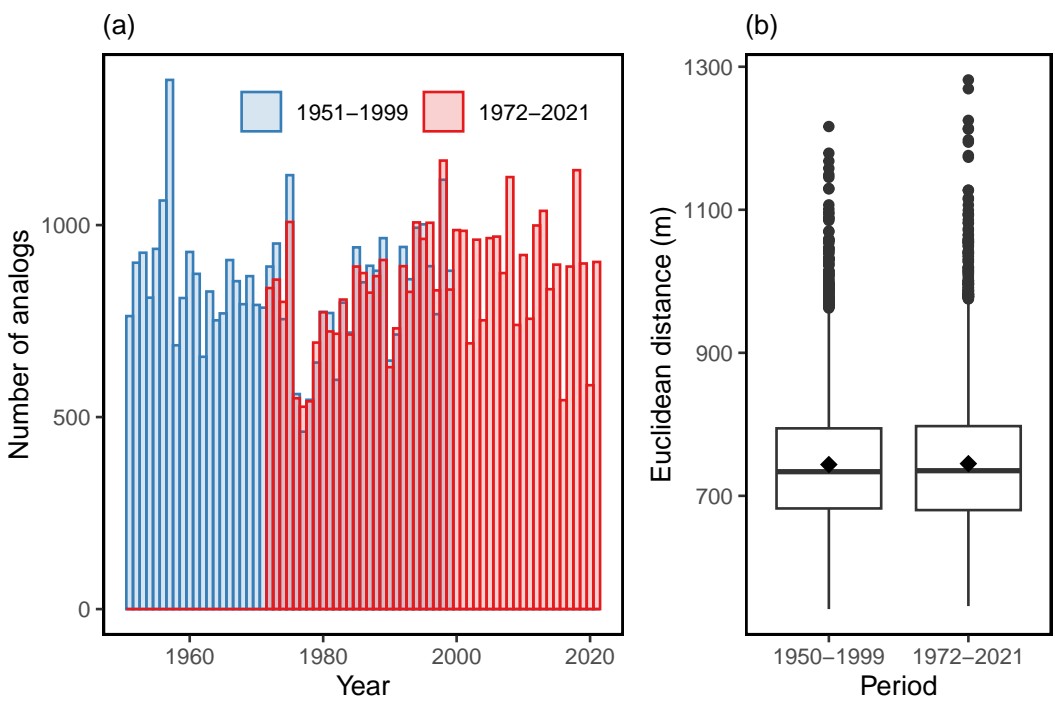

**Figure B1.** (a) Histograms of years of the $K = 20$ best analogs for each day of the 1951-2021 in October to March, in either 1951-1999 (blue) and 1972-2021 (red). (b) Euclidean distances between Z500 fields (m) of the analogs in the same two periods.

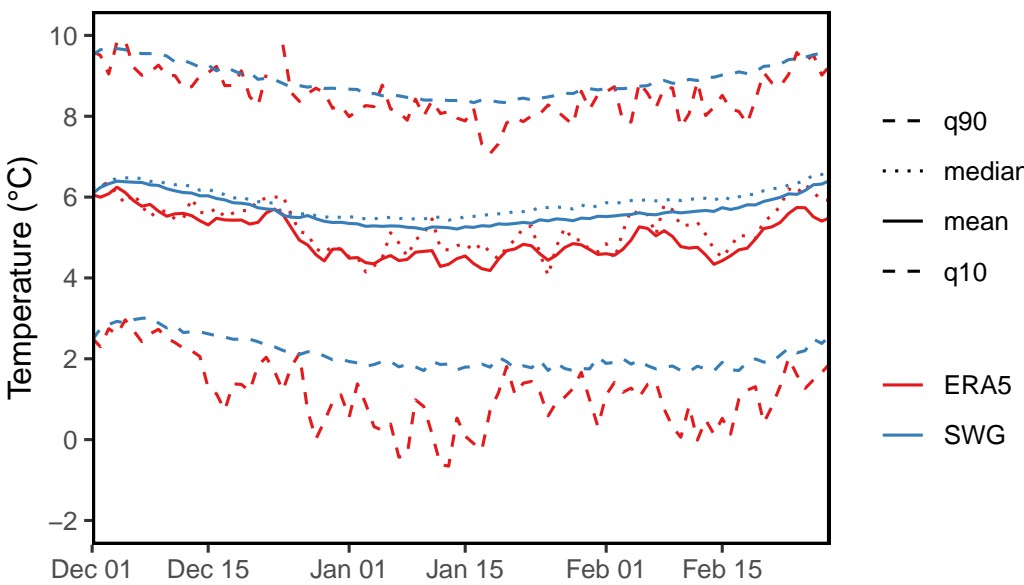

**Figure C1.** Winter seasonal cycle time series, as computed from winters from 1951 to 2021 in ERA5 (red) or from SWG simulations (blue) initialized at the start of the same winters (1000 simulations per winter). For both, the mean (solid lines), the median (dotted lines) and the $10^{th}$ and $90^{th}$ percentiles (dashed lines) are displayed.

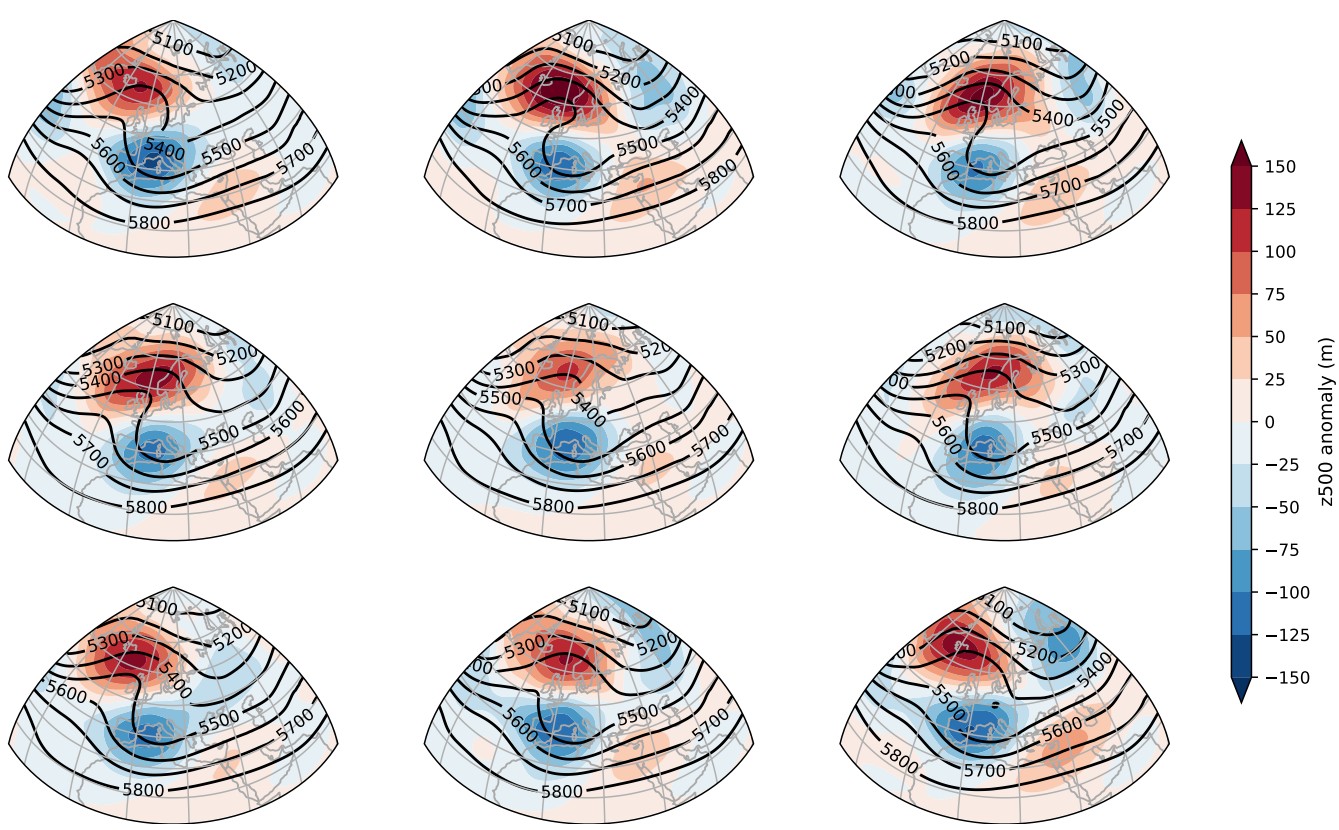

**Figure D1.** Composites of Z500 anomaly for the nine coldest individual SWG simulations made from 1972-2021 analogs. The color scales have a wider range compared to Fig. 6 as they are fitted for individual events.