# Peer review of "Simulating record-shattering cold winters of the beginning of the 21st century in France"

_EGUsphere, 2024_

## Author Comment (AC1)

**Reply to Review of "Simulating record-shattering cold winters of the beginning of the 21st century in France" (Reviewer #1)**

We thank the reviewer for their careful reading of our manuscript and their constructive remarks. Our replies are in **red**.

In the paper the authors use rare-event algorithm for sampling and simulating cold spell events in France in order to evaluate their dynamics. They use winter DJF 1963 as a reference and then used a weather generator method to simulate the coldest winter that occur in two periods with less and more effect of climate change. Their results show that the frequency of cold spells is decreasing but their intensity remains stationary and that few simulated cold spells reach the intensity of the winter 1963. They also analysed the atmospheric circulation of the observed and simulated events.

The paper is highly technical and often hard to follow, with many grammatical errors so that I suggest to have it checked by an English native speaker. Said so, the topic is very interesting and I believe that this can be published in WCD after addressing some comments below.

**Major comments**

Abstract: the conclusion of the study is missing in the abstract. I suggest to add 1-2 sentences at the end, that resume the importance of the work.

The abstract has been modified to better highlight the conclusions of the study.

L63 not clear what the bullet point states. "Reached without information..."? reached by who or what? Which information? Please clarify.

The sentence has been clarified. Reviewer 2 made a similar comment. We gave a detailed answer (to Reviewer 2) below.

L102-109 it would be helpful if you could highlight in the figure the strongest cold spells mentioned.

 OK. The coldest spells mentioned in the text are now highlighted in the figure.

If you focus on 90-day running means then why in Fig. 1b you show only 7-day running means?

With Fig. 1b, we want to show the daily temporal series of DJF for each year. This daily series is smoothed over a 7-day window to improve readability. This is clarified in the text.

**Minor comments**

L4-5 "significant computational resources"

OK. The correction has been made

L9 repetition of "France", I suggest remove it

OK.

L8 "We focus first…". OK, but I don't see the continuation of this in the remaining text. For example, "Secondly, we apply…" or "Then, we use…". I suggest remove "first" or adapt the following text accordingly.

OK. This is clarified now.

L11 "We applied..", before you said "We find..". Please check the tenses in all the Abstract and make sure to be consistent with the chosen one.

We unified the tenses in the abstract.

L31 "Arctic Amplification", please remove "The".

OK. Removed

L33 "But…" change to ", but…"

OK.

L62 "the observational periods."

OK.

L67 "200 years for simulations from the.." ???

This refers to historical and scenarios simulations of the Coupled Model Intercomparison Project Phase 6 (CMIP6), which usually run from 1850 to 2100.

L69 "2000-2050 decade" ?

We replaced by "2000-2050 period".

L74 reference missing

The missing reference has been added.

L98 "have been"

OK, we made the correction.

L120 no need to repeat "Methods:"

OK. The repetition has been removed.

Figures 2: not clear the meaning of blue and red histograms. Please add it in the caption.

The caption has been modified to clarify the meaning.

---

## Author Comment (AC2)

**Reply to Review of "Simulating record-shattering cold winters of the beginning of the 21st century in France" (Reviewer #2)**

We thank the reviewer for their careful reading of our manuscript and their constructive remarks. Our replies are in **red**.

**Major comments**

In this manuscript, the authors study extreme cold spells over France using a stochastic weather generator. The interest of the manuscript is twofold. On the one hand, there is a strong methodological motivation, as rare event studies suffer from a problem of lack of data, and the proposed method is sufficiently general to potentially be applied to many other cases. While the idea of approximating the true dynamics of the atmosphere using circulation analogues is not new, the current study combines it with an additional ingredient, the idea of biasing the statistics towards extreme events, which they refer to as importance sampling. On the other hand, the authors ask whether their tool allows to assess whether the probability of extremely cold winters such as the 1963 winter over France have been affected by climate change, which is a valid question in its own right. Hence, the motivations for this research seem robust to me, and the manuscript should be of interest to a sufficiently large audience. However, there are a number of points which I think should be clarified about the method and the validity of the results.

First I think the objectives of the paper are not framed in a very precise way, which makes it a bit difficult to assess to what extent they are reached. It is said in the introduction that the objectives are to examine: "1. whether a winter as cold as 1963 can be reached without information on this event 2. whether such a winter is still possible in the 21st century". The meaning of objective 1 is not clear a priori; if you were doing a simulation with a GCM for instance the answer would clearly be yes. So it seems this objective only makes sense within the framework of the stochastic weather generator used in the manuscript. If that is indeed the case I think the objective should be rephrased to clearly explain which property of the methodology is being tested.

The objectives of the paper have been rephrased in a more precise way. The 1962/63 winter mean temperature came as a cold outlier in the ERA5 reanalysis, even in the 20th century. We first evaluate whether a statistical model can reach such low mean temperatures given information that exclude that winter. This evaluation is based on a concept outlined by Corti et al. (Nature, 1999) that combinations of weather patterns can lead to warm (or cold) temperatures.

I understand objective 2, but the word "possible" in this sentence seems a bit surprising to me. Similarly, the manuscript refers several times to "the coldest winters possible".

This terminology has been rephrased. In spite of an increasing winter temperature trend, we evaluate the ability of an SWG to simulate such a low winter given climate information of the present decades. We will avoid expressions like 'coldest possible", as they are confusing.

From the point of view of the true statistics of the climate system, it seems unlikely that a hard threshold exists (at least not within such bounds), it is more likely just a matter of how the probability of such an event changes with global warming.

Indeed, there might be such "hard thresholds" from first principles (e.g. Röthlisberger and Papritz, 2023), but comparing them with our approach might not be relevant, as such thresholds are given for "instantaneous" temperature, not whole winter averages.

However, my impression is that the authors carefully avoid to talk about probabilities, and restrict themselves to qualitative statements. I understand that the probabilities obtained with the SWG are biased, but if I understand correctly it should be able to go back to the probabilities with respect to the unbiased distribution, and therefore the method should allow to make quantitative statements and estimate such probabilities. This, I think, makes the method quite interesting. For instance, it should allow to compare the probability or return time of an event with a given amplitude (or return level), like the 1963 winter over France, both in the factual and counterfactual worlds, and compare these two estimates. Is this correct or am I missing something?

In theory, one could derive the statistical moments of the SWG model, like in the rare event algorithm used by Ragone et al. (2018). The mathematical expectation of the SWG simulations is a discrete version of a Laplace transform of the temperature distribution of analogues because the weights are exponentials of the rank of analog temperatures). But, given the finite and rather small number of analogs, this leads to a numerically poor estimation of the moments of the SWG model. This is explained in Yiou and Jézéquel (2020). Therefore, in practice, we prefer to use a more simple and pragmatic approach, by comparing the distribution of the simulated means with a climatological ensemble.

A second, related aspect is that I am not sure I understand what is the basis for the conclusion that extremely cold winters such as the 1963 winter are not significantly affected by climate change, even in the careful terminology adopted by the authors.

As far as I understand, the conclusions are drawn on the basis of Fig. 5, which shows that the distribution of TG90d simulated from a SWG based on 50 years of data from the first part of the reanalysis dataset does not differ significantly from the one simulated based on the last 50 years of the reanalysis dataset.

By doing so, it seems to me that you are comparing two distributions which are both biased (due to the weights in the stochastic weather generator), but potentially not in the same way. Hence, I think the statistics should be unbiased before any conclusion can be made (for instance by computing the probability or return time of an event at the

level of the 1963 winter). Do you agree, or did you make sure that the two distributions were biased in the same way?

The importance sampling weights that produce the "bias" are always the same because they are based on the ranks of temperature analogs, between 1 and 20, not on the temperature values themselves. Therefore, the simulations are biased towards low temperatures in the same way in the two analog periods.

Similarly, in section 2.5 and Fig. 2 you study statistics of the time derivative of Z500 and temperature, and compare the distributions obtained directly from data, using the stochastic weather generator, or some naive random sampling. But is it clear that we expect the distributions to be the same, or should the one obtained with the SWG also be biased ?

The distribution of derivatives obtained from SWG simulations may indeed be biased due to the analogue resampling process and the specific focus on simulating extreme winters. These two biases are expected to increase the variance of time derivatives. Therefore, we aimed to examine the distribution of time derivatives to ensure that the atmospheric consistency of the SWG output events does not significantly differ from the ERA5 winter observations.

Finally, I have a number of more general questions on the methodological choices related to the SWG:

- what is the rationale for biasing using the rank of the temperature of the analogues rather their than absolute temperature value? Does it make any difference?

As discussed by Yiou and Jézéquel (2020), using the absolute value of temperature requires many scaling discussions due to the units of the variable, which disappear when using the rank, which are values between 1 and 20 (20 analogs). So there is an obvious practical consideration. In addition, if one wants to determine the expected value of the simulations, one obtains the Laplace transform of the temperature distribution (as a function of the $\alpha\_T$ parameter), which, in theory, gives access to the moments of the distribution of the simulations, in the asymptotic case of infinite analogs (as mentioned above). Thus, there is also a theoretical rationale for using ranks rather than absolute values. This will be recalled in the text, although we do not delve into those mathematical properties.

- is there a justification for the form of the biasing factor as a product of exponentials? Or is it empirical?

The importance sampling applied to the SWG is inspired by the importance sampling of a climate model as developed in Ragone et al. (2021) who used the GKLT large deviation algorithm to select trajectories with exponential weights.

We acknowledge that taking the product of weights on calendar time and temperature ranks is completely empirical. It is also the simplest practical way to proceed. A multiplication of weights is like an "AND" condition (in logic), which is what we aim for. An addition would lead to an "OR" condition, which is not what we want. Exponentials produce decreasing functions that are infinitely differentiable, and have no singularity around 0. They have the simplest functional form that achieves those properties. So, unless there is a hard reason not to use exponentials, it seems preferable to keep them.

- is there a way to choose the parameters $\alpha_{cal}$, $\alpha_T$ a priori, or is it done entirely empirically?

The selection of the parameters alpha_cal and alpha_T is made empirically as in Yiou and Jézéquel (2020). The criteria of selection are shown in Fig. 3.

- if I understand correctly, the SWG allows transitions from the current state to analogues of the image of the current state in the true dynamics. I guess it would be possible to do things differently and allow transitions from the current state to images of its analogues. Would it make any difference? Is there a good reason for making one choice or the other?

This is an interesting remark. We performed SWG simulations according to this suggestion (select a random analog of day t, then consider its next day), and found that it is not equivalent to our procedure (select a random analog of day t+1) and produces less extreme values for TG90d. The reason is (relatively) simple: if we chose the procedure suggested by the referee, there is no constraint on the derivative of the simulation, which just follows the derivative of the analogs. Our procedure favors a negative derivative (for cold temperatures), and hence constrains the trajectories to the lower temperatures. To be more explicit: in our procedure, if temperature at day t+1 is warmer than at day t, it suffices that one of the analog temperatures of day t+1 is lower than at day t to go "colder". In the procedure suggested by the referee, if one chooses a colder analog at day t, then there is a chance that its next day (t+1) is warmer. This produces a positive derivative (or increasing temperature). Therefore, we prefer to keep our procedure.

[Figure]

*Figure 1: Boxplots of the SWG outputs initialized on December 1st 1962 for 1951-1999 and 1972-2021, with transitions either from the current state to analogues of its image (blue) or to images of its analogues (red).*

- the SWG seems like an interesting way to estimate properties of rare events, as mentioned above. Here the validation of such estimates is hindered by the fact that the dataset is quite small. Has the method been validated on a longer dataset, such as model data, and if so, for which type of quantities?

The method has been tested to simulate European heatwaves from analogs of a long climate model control simulation (Miloshevich et al., 2024), with a careful verification protocol. It turns out that this approach allows estimating the tail of the distribution of extreme heatwaves.

For cold spells in Europe, Sippel et al. (2023) made a comparison with the boosting approach of Gessner et al. (2021).

**Specific comments**

In addition to the above general comments, I have a number of questions on more specific points in the manuscript.

- The manuscript introduces a quantity, TGrd, which can be characterized by different event durations $r$. I understand that the paper focuses on the case where $r$ is the

whole winter, and this is fair enough, but it would be interesting to illustrate how the distribution of TGrd changes for different values of the duration $r$. Presumably the distribution becomes closer to Gaussian at larger values of $r$, and the variance should also be reduced, as seems to be the case on Fig. 1a. A related comment is that some of the extreme events at large $r$ might be made of several extreme events at small $r$. It would be interesting to quantify this connection.

[Figure]

*Figure 2: Yearly temperature distributions of TGrd from 1950 to 2021 for various r.*

If we plot the distribution of the annual minima of TGrd for the four r values considered in the article (3, 10, 30, 90), we obtain the graph presented in Fig. 2. As anticipated, the variance of TGrd decreases with increasing r.

The distribution for r equal to 90 (meaning the distribution of yearly winter mean temperatures) indeed yields a p-value lower than 0.05 when conducting a Shapiro-Wilk normality, only the ones with r equal to 3 or 90 yield p-values lower than 0.05. As for the other scales, we compute the minimum over a yearly block, the resulting distribution should tend to a Generalized Extreme Value (GEV) distribution for lower r. However, given the non-stationnarity of the data and the relatively small sample sizes (n=71), it is challenging to draw any definitive conclusions.

| Length of extreme at smaller time scale | 3 | 10 | 30 | 90 |
|---|---|---|---|---|
| 3 | 1 | 1.53 | 2.3 | 4.12 |
| 10 | 0 | 1 | 1.55 | 2.25 |
| 30 | 0 | 0 | 1 | 1.13 |

*Table 1: Average number of cold winter extremes at smaller scale encompassed in extremes at higher scale.*

To investigate the prevalence of extreme events at a small scale within extreme events at a larger scale, we first identify the 10% coldest of the non-overlapping local minima at each time scale. Next, we determine the average number of extreme events at a smaller time scale that coincide with those at a higher time scale. The results are presented in Table 1 and can be interpreted as follows: "On average, a 90-day extreme event encompasses 4.12 3-day cold spells." This indicates that extreme events at a larger time scale are indeed composed of extremes at a smaller time scale, but only to a certain extent. For instance, out of the 8 coldest 90-day events, only winter 1963 contains two distinct 30-day extremes. The remaining events include only one short-lived cold event.

- ERA5 data is not stationary. It could seem natural to detrend it before constructing the SWG, but it seems like this was not done here. Is there a good reason for that? If the data indeed has a trend, is there anything in the SWG which prevents it from drifting, due to the biasing factor, towards earlier years in the dataset where low temperature extremes were more frequent?

We agree with the reviewer that ERA5 data is not stationary. This is shown for temperature time series at different time scales in Fig1 of the manuscript. This also appears on Z500. Thus we first checked that the non-stationarity does not affect the distribution and quality of the analogs. Fig 4a shows the year of the 20 best analogs of each winter day of 1950-2021 either in 1951-1999 or 1972-2021 while Fig 4b compares the quality of the two same sets of analogs. We see that there is no major difference between the two periods and that analog years show no meaningful trend.

[Figure]

*Figure 4: (a) Histogram of the year of the 20 best analogs of each winter day of 1950-2021 (red) and 1973-2021 (blue).*
*(b) Boxplots of the analogues quality for winter days of the 1950-1999 and 1972-2021 periods.*

Regarding the analogs selected during the SWG simulations, we agree with the reviewer that there may indeed be a minor bias towards earlier years in the dataset. However, this is not of meaningful concern. The median and mean of the selected analogue years are 1972 and 1973, respectively, for the 1951-1999 period, and 1992 and 1994, respectively, for the 1972-2021 period. Therefore, we conclude that the SWG simulations are representative of the climate period of the analogues.

- In section 2.6 and Fig. 3, you show the fraction of simulations for which the last day of the simulations falls after Feb 15. I am not sure I understand the rationale for this choice. What do you expect to be a measure of the quality of the stochastic weather generator based on this metric? Why not computing the climatology simulated by the SWG for instance?

The purpose of the alpha\_cal weights is to ensure that the simulations progress in time and adhere to the seasonal cycle, thereby generating realistic events that do not resemble an "eternal winter" scenario with persistently extreme cold temperatures. Yiou and Jézéquel (2020) developed this calendar weighting approach to achieve this objective, and the proportion of days falling at the end of the period of the event is a method they previously proposed to control how simulated events do not deviate from the seasonal cycle.

A climatology of the SWG is implicitly computed in Figure 5, when we initialize the SWG from all 1st of December in the ERA5 reanalysis with a calendar constraint but without importance sampling. Seasonal cycles are not smoothed. The figure demonstrates that the SWG reproduces a seasonal cycle, although the mean seasonal cycle is not as cold as

in ERA5. This indicates that the SWG results are conservative and that colder events may also be possible.

[Figure]

*Figure 5: 10th percentile, mean, median and 90th percentile of the 71 ERA5 daily winter temperature (red) and SWG output simaltions initialized at each start of the same 71 winters.*

- Fig. 4 shows that a SWG constructed without 1963 data simulates typically less cold winters than one including that data, and different simulations exhibit smaller variability. The first point seems like a drawback. Do I understand correctly that the benefit that you gain in return is the reduced risk of simulating minor variants of the 1963 winter, and therefore more strongly correlated events? How is this related to the smaller variability between independent runs obtained when excluding 1963? Do you know what is the fraction of days picked from 1963 in the simulations with the SWG constructed from data including that year, or do you have another measure of the degree to which they are correlated?

By permitting the selection of days from winter 1963 as in Yiou and Jézéquel (2020), the events obtained in the two periods would have an average of 36.0% and 31.5% of days picked from 1963, respectively. Consequently, these events would essentially be variations of the winter 1963 and thus more correlated with each other. With this configuration, the standard deviation between simulations in the 1951-1999 period is 0.49°C. With the new configuration, the standard deviation increases to 0.71°C, as the simulated events are less correlated with the winter 1963.

Including 1963 also did not align in an attribution perspective, and it is common practice to exclude the event of interest from the both factual and counterfactual when

comparing the two to prevent selection bias (see for instance Vautard et al., 2020, Zeder et al. 2023).

- Section 3.3 and Fig. 6; I would be curious to know whether composite maps are good representatives of individual events.

Composite maps are quite good representatives of individual events as extreme events tend to look like each other. In Fig. 6 for instance are the Z500 mean maps of the 9 coldest simulations from the SWG using analogues in 1972-2021.

[Figure]

*Figure 6: Absolute values (contours, in m) and anomalies (in m) with respect to 1950-2021 (shaded areas) of 500-hPa geopotential height (Z500) average over the simulation for the 9 coldest simulations from the SWG, using analogues in 1972-2021.*

**Technical details and typos**

- Abstract: "societal impacts on society" is redundant

OK, "on society" has been removed.

- p3, L68 "(e.g. in CMIP6 archive)" isn't the number of ensemble members missing in the parenthesis? (50?)

- p3, L74 broken reference for "Sippel et al. [REF]"

The missing reference has been added.

- p4, L104: "a upward trends" should be "an upward trend" I guess.

OK, correction made.

- p4: it would be helpful if the formula defining TGrd based on TG could be written. Similarly on p6, the definition of the stochastic weather generator, which is currently described in words, might be easier to understand if the corresponding mathematical formulas were given, or if a reference to a paper where this is done was given.

We added the formula for TG*r*d for an event of duration r:

$$TGrd_t \ = \ \frac{1}{r} \sum_{i=-r/2}^{(r/2)-1} TG_{i+1} \qquad \text{where } TG_t \text{ is the daily average temperature at time step t.}$$

The formula for an odd value of r is equivalent, but with the sum centered around r.

We detailed the formula for the weights computation but we have not added any more equations so as not to overload the paper. Additional details can be found in Yiou and Jézéquel (2020).

- p6, L124 "K analogs days" should be "K analog days"

OK, correction made.

- p6, L139 "The goal is to simulate L day trajectories of a model while optimizing an observable", I am not sure I understand what the authors mean by "optimizing an observable".

OK, replaced by "maximizing".

- p8, section 2.6 title "Simulation protocole" should be "protocol"

OK, correction made.

- p11, caption of Fig. 4: second occurrence of "vertical purple dashed line": I guess you meant "horizontal"

Indeed, "vertical" has been changed to "horizontal".

**References**

Corti, S., Molteni, F., & Palmer, T. N. (1999). Signature of recent climate change in frequencies of natural atmospheric circulation regimes. *Nature*, *398*(6730), 799–802. https://doi.org/10.1038/19745

Gessner, C., Fischer, E. M., Beyerle, U., & Knutti, R. (2021). Very Rare Heat Extremes: Quantifying and Understanding Using Ensemble Reinitialization. *Journal of Climate*, *34*(16), 6619–6634. https://doi.org/10.1175/JCLI-D-20-0916.1

Miloshevich, G., Lucente, D., Yiou, P., & Bouchet, F. (2024). Extreme heat wave sampling and prediction with analog Markov chain and comparisons with deep learning. *Environmental Data Science*, *3*, e9. https://doi.org/10.1017/eds.2024.7

Ragone, F., Wouters, J., & Bouchet, F. (2018). Computation of extreme heat waves in climate models using a large deviation algorithm. *Proceedings of the National Academy of Sciences of the United States of America*, *115*(1), 24–29. https://doi.org/10.1073/pnas.1712645115

Ragone, F., & Bouchet, F. (2021). Rare Event Algorithm Study of Extreme Warm Summers and Heatwaves Over Europe. *Geophysical Research Letters*, *48*(12). https://doi.org/10.1029/2020GL091197

Röthlisberger, M., & Papritz, L. (2023). A Global Quantification of the Physical Processes Leading to Near-Surface Cold Extremes. *Geophysical Research Letters*, *50*(5). https://doi.org/10.1029/2022GL101670

Sippel, S., Barnes, C., Cadiou, C., Fischer, E., Kew, S., Kretschmer, M., Philip, S., Shepherd, T. G., Singh, J., Vautard, R., & Yiou, P. (2023). An extreme cold Central European winter such as 1963 is unlikely but still possible despite climate change. *EGUsphere*, 1–24. https://doi.org/10.5194/egusphere-2023-2523

Vautard, R., van Aalst, M., Boucher, O., Drouin, A., Haustein, K., Kreienkamp, F., van Oldenborgh, G. J., Otto, F. E. L., Ribes, A., Robin, Y., Schneider, M., Soubeyroux, J. M., Stott, P., Seneviratne, S. I., Vogel, M. M., & Wehner, M. (2020). Human contribution to the record-breaking June and July 2019 heatwaves in Western Europe. *Environmental Research Letters*, *15*(9), 18. https://doi.org/10.1088/1748-9326/aba3d4

Yiou, P., & Jezequel, A. (2020). Simulation of extreme heat waves with empirical importance sampling. *Geoscientific Model Development*, *13*(2), 763–781. https://doi.org/10.5194/gmd-13-763-2020

Zeder, J., Sippel, S., Pasche, O. C., Engelke, S., & Fischer, E. M. (2023). *The effect of a short observational record on the statistics of temperature extremes*. https://doi.org/10.22541/essoar.168167197.74742164/v1

---

## Author Response (AR2)

**Reply to Editor comments of "Simulating record-shattering cold winters of the beginning of the 21st century in France"**

We thank the editor for their evaluation of the manuscript and their additional comments. Our replies are in **red**.

- in the first round of reviews, reviewer 1 said "the conclusion of the study is missing in the abstract. I suggest to add 1-2 sentences at the end, that resume the importance of the work." This has not been sufficiently addressed, the last sentence still does not describe what you find, where an outlook would be more fitting.

The abstract was modified to make the conclusions of the study clearer. (L14-17)

Reviewer 1 also suggested to go over the manuscript to do langugage corrections. I'm adding my suggestions here, but please note that these are not comprehensive, and you may want to go over the manuscript again to improve readability.

We thank the editor for the suggested corrections. Additional language improvements have been made along the manuscript to improve readability.

abstract: use "allow" + object

Ok, correction made. (L4)

reviewer 1 comment: "L102-109: it would be helpful if you could highlight in the figure the strongest cold spells mentioned."

The figure in the revised version looks unchanged without the highlights, please check.

Black vertical dashed lines were added to the figure to outline the events mentioned in the text.

line 67: "during which climate has been warmer"

Ok, the sentence was rephrased.

line 68: "simulating an ensemble"

Ok.

line 71: 50 ensemble members or 50 ensembles?

The sentence has been clarified to "50 ensemble members".

line 105, 111: change "time scales" to "event durations"?

Ok, we reformulated (L106-115)

line 106: minimum value of temperature for each event?

We modified the sentence for clarification. The formula refers to an even value of the duration r. (L108)

line 107: might want to explain what t is before using it, e.g. say "over all time steps t"

Ok. (L108)

line 117: "This Gaussian interpretation" -> could you explain a bit more? e.g. sth like "the assumption of … to be Gaussian"
We clarified accordingly. (L119)

line 141: "analog years"
Ok.

line 169: "avoids discussions": might want to change to sth like "allows for an interpretation independent of the scaling..."&
The sentence has been rephrased. (L172-174)

line 171: correct: "at each time step"
Ok.

line 219: "we simulated" would suggest to stay in present tense, see reviewer suggestions for abstract.
We modified the text to keep the present tense.

385: "analog quality"
Ok.

Figure C1 caption: "plain lines" -> "solid lines" also, if you write "(dashed lines)" then should also say what the dotted lines are.
Ok. We modified the caption accordingly. The meaning of dotted lines has been added.

Figure D1: "as in Fig. 6 for" ?
The caption of figure D1 has been corrected by adding instead "The color scales have a wider range compared to Fig. 6 as they are fitted for individual events".

---

## Author Response (AR3)

**Reply to co-editor corrections of "Simulating record-shattering cold winters of the beginning of the 21st century in France"**

We thank the co-editor for their evaluation of the manuscript and their additional technical corrections. Our replies are in **red**.

line 263: correct to:
"This is a consequence of the fact that configuration 2 does not use information from the observed event apart from the initial conditions."
Ok, done.

line 283: correct to: "In other words, extremely cold winters do not warm at the same rate as winter mean temperatures."
Ok, correction made.

Line 293: correct to: "We find that the likelihood of reaching such cold winter temperatures…" or
"We find that the likelihood of such cold winters …"
Ok, we changed to "We find that the likelihood of such cold winters …"

line 304: correct to: "The respective positive and negative Z500 anomalies over Iceland and the Azores are however characteristic of NAO−, which is …"
Ok, correction made.

line 361: "…when the national thermal indicator goes under under −2°C for at least three days"
correct to: "falls below"
Ok, we changed to 'falls below".